# Genetic Factors Associated with the Development of Neuropathy in Type 2 Diabetes

**DOI:** 10.3390/ijms25031815

**Published:** 2024-02-02

**Authors:** Dóra Zsuszanna Tordai, Noémi Hajdú, Ramóna Rácz, Ildikó Istenes, Magdolna Békeffy, Orsolya Erzsébet Vági, Miklós Kempler, Anna Erzsébet Körei, Bálint Tóbiás, Anett Illés, Henriett Pikó, János Pál Kósa, Kristóf Árvai, Márton Papp, Péter András Lakatos, Péter Kempler, Zsuzsanna Putz

**Affiliations:** 1Department of Internal Medicine and Oncology, Semmelweis University, 1083 Budapest, Hungary; hajdu.no3mi@gmail.com (N.H.); istildi78@gmail.com (I.I.); bekeffy.magdi95@gmail.com (M.B.); vagiorsi@gmail.com (O.E.V.); or anna.korei@yahoo.com (A.E.K.); balint.tobias@gmail.com (B.T.); illes.anett@med.semmelweis-univ.hu (A.I.); piko.henrietta@med.semmelweis-univ.hu (H.P.); janos.kosa@gmail.com (J.P.K.); lakatos.peter@med.semmelweis-univ.hu (P.A.L.); kempler.peter@med.semmelweis-univ.hu (P.K.); or zsuzsannaputz@yahoo.com (Z.P.); 2Department of Internal Medicine and Hematology, Semmelweis University, 1085 Budapest, Hungary; kemplersoma@gmail.com; 3PentaCore Laboratory, 1134 Budapest, Hungary; kristof.arvai@gmail.com; 4Vascular Diagnostics Ltd., 1026 Budapest, Hungary; 5Eötvös Lóránd Scientific Network ENDOMOLPAT, Semmelweis University, 1085 Budapest, Hungary; 6Centre for Bioinformatics, University of Veterinary Medicine, 1078 Budapest, Hungary; pappmarci95@gmail.com

**Keywords:** type 2 diabetes, neuropathy risk, genetic variants

## Abstract

Neuropathy is a serious and frequent complication of type 2 diabetes (T2DM). This study was carried out to search for genetic factors associated with the development of diabetic neuropathy by whole exome sequencing. For this study, 24 patients with long-term type 2 diabetes with neuropathy and 24 without underwent detailed neurological assessment and whole exome sequencing. Cardiovascular autonomic function was evaluated by cardiovascular reflex tests. Heart rate variability was measured by the triangle index. Sensory nerve function was estimated by Neurometer and Medoc devices. Neuropathic symptoms were characterized by the neuropathy total symptom score (NTSS). Whole exome sequencing (WES) was performed on a Thermo Ion GeneStudio S5 system determining the coding sequences of approximately 32,000 genes comprising 50 million base pairs. Variants were detected by Ion Reporter software and annotated using ANNOVAR, integrating database information from dbSNP, ClinVar, gnomAD, and OMIM. Integrative genomics viewer (IGV) was used for visualization of the mapped reads. We have identified genetic variants that were significantly associated with increased (22–49-fold) risk of neuropathy (rs2032930 and rs2032931 of *recQ-mediated genome instability protein 2* (*RMI2) gene)*, rs604349 of *myosin binding protein H like* (*MYBPHL) gene* and with reduced (0.07–0.08-fold) risk (rs917778 of *multivesicular body subunit 12B (MVB12B)* and rs2234753 of *retinoic acid X receptor alpha* (RXRA) genes). The rs2032930 showed a significant correlation with current perception thresholds measured at 5 Hz and 250 Hz for n. medianus (*p* = 0.042 and *p* = 0.003, respectively) and at 5 Hz for n. peroneus (*p* = 0.037), as well as the deep breath test (*p* = 0.022) and the NTSS (*p* = 0.023). The rs2032931 was associated with current perception thresholds (*p* = 0.003 and *p* = 0.037, respectively), deep breath test (*p* = 0.022), and NTSS (*p* = 0.023). The rs604349 correlated with values measured at 2000 (*p* = 0.049), 250 (*p* = 0.018), and 5 Hz (*p* = 0.005) for n. medianus, as well as warm perception threshold measured by Medoc device (*p* = 0.042). The rs2234753 showed correlations with a current perception threshold measured at 2000 Hz for n. medianus (*p* = 0.020), deep breath test (*p* = 0.040), and NTSS (*p* = 0.003). There was a significant relationship between rs91778 and cold perception threshold (*p* = 0.013). In our study, genetic variants have been identified that may have an impact on the risk of neuropathy developing in type 2 diabetic patients. These results could open up new opportunities for early preventive measures and might provide targets for new drug developments in the future.

## 1. Introduction

Diabetic neuropathy is a serious consequence of diabetes diagnosed in about one-third of all diabetic patients [1]. It is also a predictor of diabetic foot ulcers [2], cardiovascular morbidity [3], and mortality [4] which are aggravating complications of diabetes. The pathogenesis mechanisms of diabetic peripheral neuropathy (DPN) and cardiovascular autonomic neuropathy (CAN) have not been completely discovered. These complications are considered to be multifactorial processes where the interactions between genetic and environmental factors might play a crucial role. Thus, genetic effects may influence the natural course of diabetic neuropathy, i.e., DPN and CAN, as well.

Only limited information is available on the influence of genetic factors in the development of diabetic neuropathy. Some data [5,6,7,8,9] corroborate the possibility of genetic susceptibility. Therefore, it is extremely important to systematically search for genetic variants at the whole exome level that may shed light on the pathomechanism of diabetic neuropathy and also identify new diagnostic and/or therapeutic targets.

This study aimed to perform whole exome sequencing in type 2 diabetic patients with and without neuropathy in order to search for genetic variants that may have an impact on the risk of developing neuropathy.

## 2. Results

The characteristics of the two study groups studied are listed in Table 1. Compared to type 2 diabetic patients with neuropathy, those without were older, but no significant differences between the groups were seen in diabetes duration, BMI, HbA1c, and lipid metabolism.

The Manhattan plot showing the logistic regression results of the single nucleotide polymorphism (SNP) association analysis can be seen in Figure 1. The results of whole exome sequencing are shown in Table 2 and Table 3 [10]. We successfully identified three genetic variants (rs2032930 and rs2032931 of the RMI2 gene and rs604349 of the MYBPHL gene) which were associated with a 22–49-fold increase in the risk of developing diabetic neuropathy.

Two other variants (rs917778 of the MVB12B and rs2234753 of the RXRA genes) appeared to have a protective effect against neuropathy, reducing the risk to 0.07–0.08. Table 2 shows the location and position of the given genetic variant on the chromosomes, as well as the chance of developing neuropathy, which is expressed by the OR.

The associations of genetic variants with neuropathic symptoms, signs and parameters that showed statistical significance can be seen in Table 3. The rs2032930 variant demonstrated a significant correlation with current perception thresholds measured at 5 Hz and 250 Hz for n. medianus (*p* = 0.042 and *p* = 0.003, respectively) and at 5 Hz for n. peroneus (*p* = 0.037), as well as the deep breath test (*p* = 0.022) and the NTSS (*p* = 0.023).

The rs2032931variant was associated with a current perception threshold measured at 250 Hz for n. medianus (*p* = 0.003) and at 5 Hz (*p* = 0.037) for n. peroneus, as well as the deep breath test (*p* = 0.022) and the NTSS (*p* = 0.023). There were correlations between rs604349 genotypes and values measured at 2000 Hz (*p* = 0.049), 250 Hz (*p* = 0.018), and 5 Hz (*p* = 0.005) for n. medianus, and the warm perception threshold measured by Medoc device (*p* = 0.042).

The rs2234753 variant was significantly related to current perception threshold measured at 2000 Hz for n. medianus (*p* = 0.020), the deep breath test (*p* = 0.04), and the NTSS (*p* = 0.003). We also identified a correlation between rs91778 and the cold perception threshold (*p* = 0.013).

Table 3 also demonstrates that the presence of rs2032930/rs2032931 variants is associated with an increase in the risk of developing both DPN and CAN. It also shows that rs604349 variant is correlated only with sensory neuropathy, while rs917778 appears to decrease the risk of sensory neuropathy. The rs2234753 variant reduces both DPN and CAN risk.

## 3. Discussion

We successfully identified genetic variants that might alter the risk of developing diabetic neuropathy. rs604349 is an intronic SNP in the MYBPHL (myosin binding protein H like) gene that seems to aggravate the risk of neuropathy. This gene has been linked to the circulation of progranulin. Progranulin is a precursor protein, expressed by many cell types throughout the body, particularly in the skin, gastrointestinal tract, and reproductive system [17], and has a complex biological function. In its full-length form, it has an anti-inflammatory effect, but after proteolytic cleavage, it promotes inflammation [18]. It is involved in angiogenesis, tumorigenesis, wound repair, cell proliferation, inflammation, and neurodegenerative and metabolic diseases [19]. Progranulin is suggested to be a marker of chronic inflammation in obesity and type 2 diabetes through adipose tissue macrophage infiltration. TNF-alfa and IL-6 are secreted by macrophages, which cause inflammation [18]. Progranulin has been proven to influence plasma lipoprotein metabolism, increasing the risk of atherosclerosis and chronic inflammatory status [20,21,22]. Patients with type 2 diabetes who have visceral obesity were shown to have a 1.4-fold increase in serum progranulin levels [18].

Progranulin concentrations were positively correlated with body mass index (BMI), fat mass, fasting glucose and insulin levels, as well as insulin resistance [23]. The granulin gene (GRN) which encodes progranulin is found in chromosome 17q. Genetic variants in the GRN gene may result in decreased serum progranulin levels [24]. Nevertheless, marked variations in circulating progranulin concentrations can be seen in wild-type GRN carriers as well. A former genome-wide study in 533 subjects found a correlation between variations in the CELSR2/PSRC1/MYBPHL/SORT1 loci on chromosome 1p and serum progranulin values [25]. Tönjes et al. [26] confirmed that the CELSR2/PSRC1/MYBPHL/SORT1 loci containing polymorphisms are associated with progranulin levels, with rs660240 being the most impactful one. The SNP found in our study might also cause high progranulin levels, resulting in increased insulin resistance, and finally hyperglycemia and chronic inflammation. As a consequence, alternative metabolic pathways might be activated in cells, such as neurons, and chronic inflammation may play a role in neuron degeneration.

rs2032930/rs2032931 are intronic SNPs found in the RMI2 (recQ-mediated genome instability protein 2) gene and appeared to increase the risk of developing neuropathy. In the literature, they have been associated with genome stability, tumorigenesis, and tumor progression [27]. In the literature, we found no clue to an existing relationship between the RMI2 gene and developing neuropathy.

In our study, rs917778 and rs2234753 were accompanied by reduced risk of diabetic neuropathy. rs917778 is also an intronic SNP in the MVB12B (multivesicular body subunit 12B) gene. This gene encodes a protein which is a member of a heterotetramer (TSG101 (Vps23), Vps28, Vps37 and MVB12A/B) called the endosomal sorting complex required for transport-I (ESCRT-I). The “task” of ESCRT is to help the generation of multivesicular organelles by clustering ubiquitinated proteins. The common function of the MVB12A/B proteins is to stimulate the downregulation of epidermal growth factor (EGF) receptors [28]. EGF is a mitogenic factor which stimulates cell differentiation. EGF also supports the differentiation, maturation, and survival of neurons [29]. Perez et al. [30] found a protective effect of EGF in acrylamide-induced neuropathy in animal experiments. This supports previous studies indicating a neuroprotective effect of EGF. The mutation of MVB12B might contribute to the longer neuroprotective effect of EGF.

Another genetic variant with reduced risk of diabetic neuropathy is rs2234753. It is also an intronic SNP in the RXRA (retinoic acid X receptor alpha) gene. RXRA is a part of the nuclear receptor superfamily and works as a transcription factor by binding homo- and heterodimers in the promoter of target genes. It plays a role in lipid metabolism, cell differentiation, and cell death [31,32,33]. In obese mouse models, RXR agonists decrease food intake and weight gain and thus maintain the balance between blood glucose and insulin sensitivity [34]. In an animal study with ducks, the RXRA gene was found to facilitate pre-adipocyte fat accumulation and differentiation via the RXRA-C/EBPA pathway. The SNP of RXRA showed a relationship with the average daily feed intake, residual feed intake, and feed conversion ratio, suggesting that RXRA is a strong metabolic regulator [35].

PPARs influence cell differentiation and regulate a number of metabolic processes involving glucose and lipid metabolism. The effects of PPARs include RXR, the vitamin D3 receptor, as well as steroid hormone receptors [36]. RXR antagonists were found to be promising therapeutical options in the treatment of T2DM because PPAR-γ with RXRs lead to the regulation of glucose metabolism [37,38]. RXRA also activates the PI3K/AKT pathway, which is known to be involved in the development of obesity and T2DM [39]. Insulin is one of the most important ligands for the PI3K/AKT pathway. The PI3K/AKT signaling pathway (reduces gluconeogenesis in liver and muscle, increases body lipid accumulation, increases glucose utilization, increase insulin secretion, and reduce appetite) is required for normal metabolism. In conditions such as excessive energy intake, it increases the circulation of free fatty acids, which destroy beta cell function and causes insulin resistance. PI3K/AKT signaling pathways are considered to be promising therapeutic targets for the treatment of the type 2 diabetes [39].

The rs2234753 SNP of the RXRA gene may have a positive effect on glucose metabolism, reducing insulin resistance; consequently, it mitigates the development of microvascular complications, such as neuropathy.

There are only a few studies that have investigated the genetic background of neuropathy; however, they analyzed only a few SNPs. These studies identified some genes [8,40] that may play a role in the development of neuropathy, including ACE, MHTFR, GST, GLO1, APOE, TCF7L2, VEGF, IL4, GPX1, ENOS, ADRA2B, ALR2, GPx-1, CAT, MIR27a, MIR499a, MIR146a, and MIR128a [41,42,43,44,45,46,47,48,49,50,51,52,53,54,55,56,57,58,59,60,61,62,63,64,65,66,67,68,69,70].

Ziegler et al. [5] investigated the role of transketolase (TKT) genetic variability in the development of neuropathy. They included 165 type 1 and 373 type 2 diabetic patients in that study. Altogether, 13 SNPs were evaluated in the TKT gene, and several correlations were found between the SNPs and peripheral nerve functions. However, most of these correlations lost their significance after Bonferroni correction, except for the correlations between the rs7648309 SNP and the symptom score, as well as rs63355988 and the feeling of warmth. Their study was the first to demonstrate an association between diabetic neuropathy and some TKT SNPs. The results suggest that TKT may play a protective role in the prevention of diabetic neuropathy.

The glyoxalase system has also been thought to play a role in the development of diabetic complications. The glyoxalase system, including glyoxalase 1 (Glo1), glyoxalase 2 (Glo2), and glutathione, prevents the production of advanced glycation end products. In particular, genetic variants of the Glo1 gene can cause changes in the structure of the glyoxalase binding site. Peculis et al. [6] were the first to document the association between rs1130534 and rs1049346 SNPs and reduced Glo1 enzyme activity.

Sleczkowska et al. [71] investigated the role of ion channels in painful diabetic neuropathy. They analyzed Kv, TRP, ANO, and HCN ion channel genes that are expressed in peripheral nerves. They used single molecule inversion probes and NGS. They found that missense heterozygous variants in the ANO3 and HCN1 genes and TRPA1 loss of function are linked to increased pain sensitivity. They also demonstrated that variations in TRPV1 and TRPV4 genes that lead to a loss of function might be present in painless diabetic neuropathy.

In another study, the role of potentially pathogenic SCG genetic variants in painful and painless diabetic neuropathy as well as in painful and painless idiopathic neuropathy was investigated [72]. They profiled 1125 patients (237 painful and 309 painless diabetic neuropathy, 547 painful small fiber neuropathy, and 32 painless single fiber neuropathy) with a single molecule inversion probe and NGS. They found an association between gain-of-function mutations in SCN9A, SCN10A, and SCN11A genes and neuron hyperexcitability, which causes pain.

In a case study of a male patient with painful diabetic neuropathy, an aspartic acid-asparagine mutation (D109N) was reported in the beta-2 subunit of the Nav1.7 voltage-dependent sodium channel, which caused hyperexcitability in the dorsal ganglion neurons [7].

In a review article, Jankovic et al. [70] summarize the most relevant gene polymorphisms of ALR2, ACE, APOE, MTHFR, NOS3, VEGF, GPx-1, and CAT genes and the epigenomics (DNA methylation, miRNA, long non-coding RNA, and post-translational histone modifications) of diabetic neuropathy.

A multicenter study involving a genome-wide approach examined one million SNPs and found a single region on chromosome 8p21.3 that was associated with neuropathy [9]. However, in this study, the existence of neuropathy was merely based on whether the patient was on neuropathy medication or not and whether the monofilament test was abnormal. In the genome locus found, 9 SNPs showed a significant correlation. These SNPs were intergenic SNPs adjacent to the glia cell line-derived neurotrophic factor (GDNF) family receptor alpha-2 (GFRA2) and the neurturin receptor gene.

All five SNPs that were identified to interfere with the risk of diabetic neuropathy in our study can be found in an intronic region of the genes, i.e., they do not get transcribed. Nevertheless, these variants might be part of higher level regulating systems that indirectly influence pathophysiological processes that may affect the development of neuropathy.

The identification of genetic variants that influence the risk of developing neuropathy has a number of implications. It may provide a basis for screening tests identifying at-risk patients early. Also, preventive measures may be applied for these patients, such as reducing cardiovascular risk (cessation of smoking, losing weight, decreasing lipid levels, etc.). Furthermore, available therapeutic interventions should be introduced as early as possible in diabetic patients at higher risk. Nevertheless, the identified genetic variants may later become therapeutic targets, as well. As an example of this, Miyashita et al. [73] described potential therapeutic interventions for neuropathy by manipulating genes such as insulin, GLP-1, PTEN, HSP27, RAGE, CWC22, and DUSP1 via the phosphatidylinositol-3 kinase/phosphorylated protein kinase B [PI3/pAkt] signaling pathway by utilizing oligonucleotids. In order to introduce our findings into routine clinical work, our results require further confirmation, including the validation of the effects of these SNPs in larger populations as well as in type 1 diabetic patients.

The limitations of our study include the relatively low number of patients; however, significant results have been demonstrated. There was a slight difference in the average age of the two groups; nevertheless, there was a considerable observation period (10.3 ± 6.2 years) in the neuropathy group and an even longer one (13.2 ± 7.5 years) in the group with no neuropathy, which makes it highly unlikely that this small difference in age could have acted as a bias. Epigenetic factors may also confound the data; however, the identical clinical characteristics of our study groups most likely eliminates this concern. In addition, the results obtained in our study are related to type 2 diabetic patients only, and no conclusion can be drawn for type 1 disease. Among the advantages, we emphasize that this is the first study to conduct an in-depth whole exome sequencing approach in an attempt to identify genetic factors impacting the development of diabetic neuropathy.

## 4. Materials and Methods

Individuals diagnosed with type 2 diabetes in primary care during a screening program were referred to our department and recruited for the present study. No healthy volunteers or type 1 diabetic patients were involved in order to ensure genetic homogeneity as is usual in genetic studies. There were 48 participants (30 men, 18 women) included who had type 2 diabetes, 24 with neuropathy and 24 without. Inclusion criteria for entry into the study were the presence of type 2 diabetes for more than 5 years, well balanced carbohydrate metabolism, and age between 18 and 69 years of age at baseline assessment. Exclusion criteria for the present study were type 1 diabetes, unbalanced carbohydrate metabolism, pregnancy, severe diseases (e.g., cancer), psychiatric disorders, immunosuppressive therapy, limited cooperation ability, and neuropathy from causes other than diabetes. The study protocol was approved by the local ethics committee (number: 37596-8/2018/EÜIG), and all participants gave written informed consent.

### 4.1. Neurological Assessment

Study subjects had a detailed neurological work-up to exclude carpal tunnel syndrome and to look for symptoms and signs of neurological impairment. Medical examination was performed paying special attention to muscle atrophy and skin alterations. Sensory nerve function was examined by the Neurometer R device (Baltimore, MD, USA). Current perception threshold (CPT) was estimated at median and peroneal nerves using Neurometer equipment at three frequencies (2000 Hz, 250 Hz, 5 Hz) assessing large and small myelinated as well as small unmyelinated sensory nerve fiber function, respectively [12,13,14]. For the three frequency measurements of the peroneal and the median nerves, the normal values were established by Evans et al. [16]. The detection of cold and heat thresholds was based on the use of a thermal sensory analyzer (TSA-II). Vibration perception threshold (VPT) was measured by a vibratory sensory analyzer (VSA-3000) on the Medoc device (Medoc Ltd., Ramat Yishai, Israel). Sensory nerve dysfunction was considered if at least two abnormal measures were found on the upper or lower limbs at any frequencies. The severity of neuropathy signs was estimated by the Neuropathy Total Symptom Score (NTSS6). The NTSS6 questionnaire measures the frequency and intensity of individual neuropathy sensory symptoms identified frequently by patients with DPN (aching pain and/or tightness; sharp, shooting, lancinating pain; and allodynia and/or hyperalgesia; numbness and/or insensitivity; prickling and/or tingling sensation; burning sensation) [11].

Cardiovascular autonomic neuropathy was evaluated through five cardiovascular reflex tests: response of heart rate to deep breathing (beat to beat variation), standing (30/15 ratio) and Valsalva maneuver (Valsalva ratio) assessing parasympathetic function, as well as blood pressure responses to standing and sustained handgrip estimating sympathetic function. The blood pressure response to sustained handgrip is no longer considered (Toronto Consensus Panel) as an acceptable clinical test but only as an investigational one [74]. Cardiosys H-01 12-lead portable ECG was utilized for all reflex tests. Cardiovascular autonomic neuropathy was diagnosed if at least one abnormal cardiovascular reflex test was present [14,15]. The normal values of Neurometer and cardiovascular reflex tests are shown in Table 4 and Table 5.

### 4.2. Genetic Analysis

#### 4.2.1. DNA Isolation

Genomic DNA was isolated from peripheral blood for the assay. This was carried out by Roche HighPure DNA Isolation Kit (Roche, Rotkreutz, Switzerland) according to the manufacturer’s instructions. The amount of isolated DNA was measured with the Qubit dsDNA HS Assay Kit (Thermo Fisher Scientific, Walzham, MA, USA).

#### 4.2.2. Whole Exome Sequencing (WES)

The exome library preparation was carried out using an Ion Torrent AmpliSeq RDY Exome Kit (Thermo Fisher Scientific, Walzham, MA, USA) following the manufacturer’s protocol. In brief, WES target regions were amplified using the AmpliSeq RDY Exome Kit and 100 ng of genomic DNA quantified by Qubit DNA HS or BR assay (Life Technologies). Then, samples were ligated to specific Ion Xpress Barcode Adapters. Libraries were amplified and purified by AMPure XP reagent to remove unamplified elements. These amplified and purified libraries were then quantified on a Qubit™ 2.0 fluorometer instrument. After that, they were diluted to ∼100 pM final concentration. Three barcoded exome libraries were handled together and combined to prepare templates which were placed on a single Ion 540 chip, and the sequencing run was performed on a Thermo Ion GeneStudio S5 system (Thermo Fisher Scientific) as instructed by the manufacturer.

The resulting AmpliSeq Exome files were aligned with human reference genome hg38. Coverage analysis was run using Ion Torrent software –v5.12-. The AmpliSeq Exome files (.BAM files) were uploaded to cloud-based Ion Reporter software -v5.12- from the Ion Torrent server. The raw data were evaluated by utilizing Ion Reporter™ software equipped with AmpliSeq Exome workflow. The variant annotation was executed by applying a number of pipelines, such as variant type SNV filter, indel, synonymous variant effect, missense, functional SIFT scores and/or PolyPhen and/or Grantham, homopolymer length ≤ 6, homozygosity, allele read count ≥ 100, allele ratio = 1.0, and minor allele frequency ≤ 0.5.

#### 4.2.3. Bioinformatic and Statistical Methods

Variants were annotated using ANNOVAR software –v03dec2019- (integrating database information from dbSNP, ClinVar, gnomAD, and OMIM. Integrative genomics viewer (IGV) was used for visualization of the mapped reads. Duplicate reads were marked using Picard. Single nucleotide polymorphisms (SNPs) were called with GATK Variant Call Format (VCF) files which were merged with BCFtools and then annotated with SnpSift [75]. The reference database used for the variant annotation was the dbSNP hg38 build 151, downloaded from the NCBI dbSNP database.

PLINK v1.9 [76] was used for the quality control of the raw VCF files. Predicted sex of the samples based on the SNP data was compared to the phenotypic sex of the individuals. SNPs were filtered based on missingness rate, minor allele frequency, and Hardy–Weinberg exact test *p*-values with thresholds of 0.05, 0.01, 1 × 10^−10^, respectively. Quality thresholds were selected based on the recommendations of [77] and the PLINK 1.9 documentation.

Association testing was performed in R v4.0.3 [78]. The GENESIS R Bioconductor package [79] was used for the logistic regression model fitting. To eliminate potential confounding effects, and due to the possible relatedness found by our quality control analysis, we controlled the model estimates for age, sex, and relatedness. Correction for the latter was performed with a genetic relationship matrix (GRM), created with the package SNPRelate [80]. A Quantile–Quantile plot was generated using the qqman R package [81] and a Manhattan plot was generated with ggplot2 [82]. The top 4 SNPs are presented based on logistic regression *p*-values.

All statistical computations were performed in R v4.0.3 [78]. Continuous variables are presented as means and standard deviations. Statistical significances were evaluated with Mann–Whitney tests. Categorical variables are given as frequencies, where the differences were evaluated with Fisher’s exact tests. A test result was considered to be significant if the *p*-value was less than 0.05.

## 5. Conclusions

In conclusion, this study demonstrates associations of genetic variants with sensory nerve fiber functions and cardiovascular autonomic neuropathy. Once our data are further corroborated, we might be able to establish new strategies for early preventative intervention and identify targets for new drug developments in the future.

## Figures and Tables

**Figure 1 ijms-25-01815-f001:**
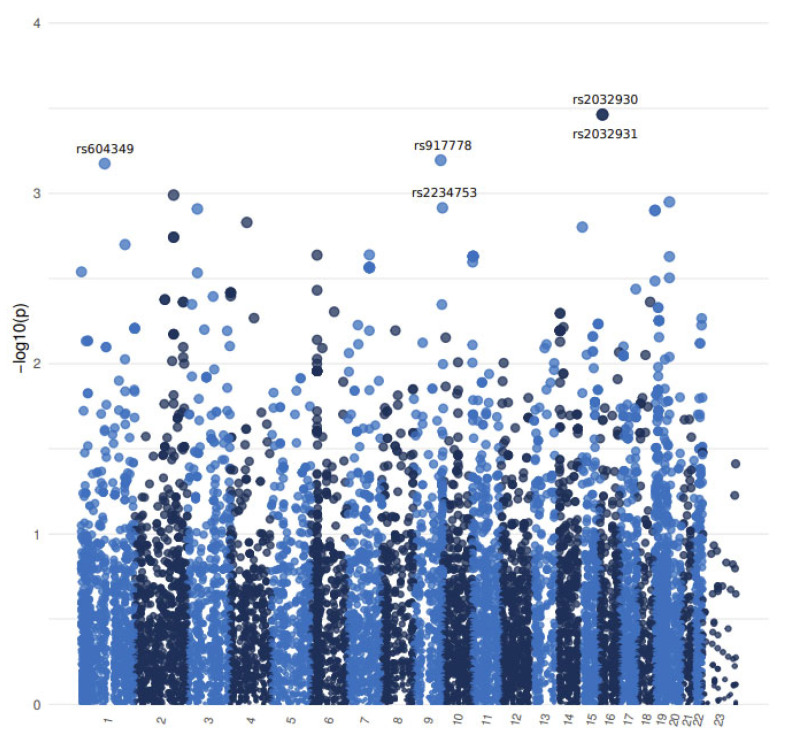
Manhattan plot showing the logistic regression results of the association analysis. SNP positions with the corresponding chromosomes are shown on the *x* axis. *p*-value as calculated by the logistic regression analysis are presented on the *y*-axis after −log10 transformation. The SNPs that were considered in this study are denoted by their rsIDs.

**Table 1 ijms-25-01815-t001:** Demographic and clinical characteristics of the two study groups.

	T2DM with Neuropathy (*n* = 24)		T2DM without Neuropathy (*n* = 24)		
	Average	±SD	Average	±SD	*p* Value
Age (years)	66.5	9.27	56.2	10.8	0.0012
Body mass (kg)	93.8	15.8	86.8	17.4	0.1009
Body height (cm)	172.5	9.9	170.0	10.5	0.4030
BMI (kg/m^2^)	31.5	5.00	30.0	5.2	0.1670
Systolic blood pressure (Hgmm)	137.7	15.7	134.0	12.2	0.3842
Diastolic blood pressure (Hgmm)	71.3	7.1	75.0	9.1	0.1718
Duration of diabetes (years)	10.3	6.2	13.2	7.5	0.1322
Sex (male/female)	17/7		13/11		
Fasting blood sugar (mmol/L)	8.92	2.81	8.97	3.18	0.9912
HbA1c (%)	7.49	1.09	7.04	1.00	0.1376
Cholesterol (mmol/L)	4.80	0.88	5.05	1.23	0.5596
LDL cholesterol (mmol/L)	2.91	0.82	3.2	0.94	0.4480
HDL cholesterol (mmol/L)	1.26	0.36	1.17	0.29	0.6441
Triglyceride (mmol/L)	1.85	0.88	2.53	1.73	0.3065

Data are reported as mean ± SD or median [IQR]. Between group differences are reported according to χ^2^ test.

**Table 2 ijms-25-01815-t002:** The results of the whole exome sequencing.

Variant ID	Alleles Minor/Major	Position	Gene	Minor Allele Frequency (MAF) of European Population *	Minor Allele Frequency (MAF) of Diabetic Patients with No Neuropathy	Minor Allele Frequency (MAF) of Diabetic Patients with Neuropathy	Logistic Regression Estimate (*β*)	Logistic Regression Estimate (*β*) Standard Error	OR for Minor Allele	*p* Value No Neuropathy vs. Neuropathy
rs2032930	T/G	chr16:11350573 (GRCh38.p14)	RMI2	0.192	0.021	0.250	3.101	0.866	22.2	0.0003
rs2032931	C/T	chr16:11350612 (GRCh38.p14)	RMI2	0.191	0.021	0.250	3.101	0.866	22.2	0.0003
rs604349	A/G	chr1:109296770 (GRCh38.p14)	MYBPHL	0.095	0.000	0.146	3.885	1.142	48.6	0.0007
rs917778	A/G	chr9:126380963 (GRCh38.p14)	MVB12B	0.206	0.313	0.104	−2.618	0.767	0.07	0.0006
rs2234753	G/A	chr9:134401915 (GRCh38.p14)	RXRA	0.246	0.292	0.167	−2.500	0.770	0.08	0.0012

The table shows the location and position of the given genetic variant on the chromosomes and minor allele frequencies, as well as the chance of developing neuropathy which is determined by the OR. *: European population allele frequencies based on the ALFA Allele Frequency Aggregator project [10].

**Table 3 ijms-25-01815-t003:** The correlations of genetic variants with neuropathic symptoms and signs as well as neurophysiological parameters in the neuropathy group.

	SNP	Minor Allele Present	Minor Allele not Present	*p*-Value
		Mean ± SD	Mean ± SD	
Current perception threshold n. medianus (mm/s)				
2000 Hz	rs604349	448.4 ± 61.5	364.2 ± 91.7	0.0491
	rs2234753	333.2 ± 47.2	376.5 ± 92.8	0.0200
250 Hz	rs2032930	175.7 ± 49.8	124.7 ± 47.8	0.0035
	rs2032931	175.7 ± 49.8	124.7 ± 47.8	0.0035
	rs604349	180 ± 14.2	132.8 ± 53.4	0.0180
5 Hz	rs2032930	98.1 ± 42.3	74.6 ± 32.5	0.0428
	rs604349	122.4 ± 28.4	78.7 ± 28.5	0.0054
Current perception threshold n. peroneus (mm/s)				
5 Hz	rs2032930	238.4 ± 270.6	169.8 ± 182.2	0.0376
	rs2032931	238.4 ± 270.6	169.8 ± 182.2	0.0376
Beat-to-beat variation (bpm)	rs2032930	8.5 ± 5.8	12.2 ± 6.6	0.0226
	rs2032931	8.5 ± 5.8	12.2 ± 6.6	0.0226
	rs2234753	13.4 ± 6.8	10.2 ± 6.3	0.0400
NTSS6 (score)	rs2032930	1.3 ± 0.5	1.7 ± 0.5	0.0230
	rs2032931	1.3 ± 0.5	1.7 ± 0.5	0.0230
	rs917778	1.7 ± 0.5	1.4 ± 0.5	NS
	rs2234753	1.8 ± 0.4	1.4 ± 0.5	0.0030
Cold detection threshold (degrees Celsius)	rs604349	39.3 ± 4.2	35.8 ± 2.6	0.0419
Heat detection threshold (degrees Celsius)	rs917778	30.2 ± 1.2	28.7 ± 3.1	0.0137

Neuropathic symptoms and signs are represented by NTSS6 (see Section 4) [11]. Neurophysiological parameters originate from the following measurements: Neurometer R device (current perception threshold), Thermal sensory analyzer, Cardiosys (five cardiovascular reflex tests) (see Section 4) [12,13,14,15,16].

**Table 4 ijms-25-01815-t004:** The internationally accepted normal values with Neurometer devices (100 = 1 mAmp) [16].

Current Perception Threshold (Frequency)	Nervus Medianus (Normal Range in mm/s)	Nervus Peroneus (Normal Range in mm/s)
2000 Hz	120–398	179–523
250 Hz	22–189	44–208
5 Hz	16–101	18–170

**Table 5 ijms-25-01815-t005:** Normal values for cardiovascular reflex testing [15].

Method	Tested Parameter	Normal Value	Borderline Value	Abnormal Value
Tests for the investigation of parasympathetic functions				
1. Deep breathing test	Beat to beat variation (beats/min)	≥15	11–14	≤10
2. Valsalva maneuver	Valsalva ratio	≥1.21	1.11–1.2	≤1.1
3. Heart rate response to standing	30/15 ratio	≥1.04	1.01–1.03	≤1.0
Tests for the investigation of sympathetic functions				
1. Blood pressure (BP) response to standing	Reduction of systolic BP (mmHg)	≤10	11–29	≥30
2. Handgrip test	Increase of diastolic BP (mmHg)	≥16	11–15	≤10

## Data Availability

https://www.ncbi.nlm.nih.gov/biosample/.

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
