# Peer review of "Genetic Factors Associated with the Development of Neuropathy in Type 2 Diabetes"

_ijms, 2024, doi:10.3390/ijms25031815_

Round 1

Reviewer 1 Report

Comments and Suggestions for Authors

I thank the editor for the opportunity to review the manuscript entitled Genetic factors associated with the development of diabetic 2 neuropathy, which offers new genetic insides in this particular complication of diabetes. I would recommend to consider the following issues:

1.       Abstract- please consider in the objective section to include the general context for your research

2.       Methodological issue: no healthy volunteers. Please add additional information for not including them

3.       Introduction: please provide an adequate explanation for not including type 1 diabetic patients in the study

4.       Consistent English editing is needed, for the scientific soundness of the manuscript. I suggest to consider a professional English editing service. (i.e. …were somewhat older but not…-line 64, the most important results…- line 69)

5.       Results section: please provide your results in an objective and impersonal way… Sentences like XYZ seemed to have or might have etc… a role in…, are not adequate for the results section. (i.e. (but not only) line 94)

6.       Please add in the limitations section the exclusion of type 1 diabetes

Comments on the Quality of English Language

1.       Consistent English editing is needed, for the scientific soundness of the manuscript. I suggest to consider a professional English editing service. (i.e. …were somewhat older but not…-line 64, the most important results…- line 69, etc)

Author Response

Dear Reviewer 1,

We thank you for considering our work for publication. We are very grateful for their valuable suggenstions. Our detailed answers are the following:

Abstract- please consider in the objective section to include the general context for your research.

Thank you for the suggestion, we have included the general context in the abstract.

Methodological issue: no healthy volunteers. Please add additional information for not including them

Thank you for this remark. We have included the following sentence: „No healthy volunteers or type 1 diabetic patients were involved in order to ensure genetic homogeneity as it is usual in genetic studies.”

Introduction: please provide an adequate explanation for not including type 1 diabetic patients in the study

Thank you for this question. Type 1 diabetes is a different disease from type 2, therefore, it might have different genetics. We wanted to make sure not to mix the two conditions to avoid the mix up of the genetics of the two diseases. Nevertheless, you are right, type 1 diabetes should be studied the same way which is being done. The sentence inserted in the Methods section explains this („No healthy volunteers or type 1 diabetic patients were involved in order to ensure genetic homogeneity as it is usual in genetic studies.”).

Consistent English editing is needed, for the scientific soundness of the manuscript. I suggest to consider a professional English editing service. (i.e. …were somewhat older but not…-line 64, the most important results…- line 69)

You are absolutely right. We have edited the text to improve scientific soundness of the manuscript.

Results section: please provide your results in an objective and impersonal way… Sentences like XYZ seemed to have or might have etc… a role in…, are not adequate for the results section. (i.e. (but not only) line 94)

Again, thank you for this criticism. We have modified the text to provide our results in an objective and impersonal way.

Please add in the limitations section the exclusion of type 1 diabetes

We added this remark to the limitations section.

Comments on the Quality of English Language

Consistent English editing is needed, for the scientific soundness of the manuscript. I suggest to consider a professional English editing service. (i.e. …were somewhat older but not…-line 64, the most important results…- line 69, etc)

We have edited the manuscript accordingly.

Again, we appreciate your help that has made our paper much more precise.

Yours sincerely,

Dr. Dora Tordai

Reviewer 2 Report

Comments and Suggestions for Authors

In the study by Tordai et al., , 24 patients with long-term type 2 diabetes and neuropathy underwent neurological assessment and whole exome sequencing. Genetic variants associated with increased or reduced risk of neuropathy were identified, suggesting potential opportunities for early preventive measures and targets for future drug developments in type 2 diabetic patients.

Introduction section:

·         The introduction provides a clear overview of the importance and relevance of diabetic neuropathy, emphasizing its prevalence and significant implications for diabetic patients.

Results section:

·         well-structured and provides a clear presentation of the findings

·         Consider incorporating visual aids like graphs or figures to enhance the presentation of complex data.

·         Clarify the meaning of " neuropathic symptoms, signs, and parameters " mentioned in Table 3 for better understanding. It might be helpful to clarify what parameters specifically refer to.

·         While the genetic variants associated with an increased or decreased risk are well-described, consider adding a brief interpretation or discussion of the clinical implications of these findings. How might these genetic associations inform future diagnostic or therapeutic strategies?

Discussion Section:

·         Provide a more detailed interpretation of the functional role of genetic variants and their potential contribution to the pathophysiology of diabetic neuropathy.

·         Explicitly discuss the clinical implications of the findings to help to understand the broader significance of the study in a clinical context. How might the identification of these genetic variants impact the prognosis, diagnosis, or treatment of diabetic neuropathy?

·         Offer a more detailed comparison with previous studies to highlight similarities or differences in the results.

·         Acknowledge and discuss any limitations of the study. Are there potential confounding factors or aspects of the research design that could affect the interpretation of results?

·         Finale, suggest potential avenues for future research based on the current findings. Are there specific aspects that warrant further investigation, and how might future studies build upon the current work?

Methods Section:

·         Specify the exact criteria used to define neuropathy and include information on normal values or reference ranges for neurological assessments.

·         Elaborate on the thresholds for quality control measures in PLINK v1.9 for transparency.

·         Provide additional details on the logistic regression model, explaining why certain variables (age, sex, relatedness) were chosen for adjustment.

·         Specify significance levels for Mann-Whitney tests and Fisher’s exact tests used for hypothesis testing.

  • General Comment:

    The cited literature appears to be not up to date, and a further check for more recent literature is advisable. Ensuring that the most current and relevant research is included will strengthen the manuscript's foundation and contextualize the findings within the latest advancements in the field.

Author Response

Dear Reviewer 2,

We are absolutely grateful for your suggestions. The corrections made based on these recommendations have improved our manuscript considerably.

In the study by Tordai et al., , 24 patients with long-term type 2 diabetes and neuropathy underwent neurological assessment and whole exome sequencing. Genetic variants associated with increased or reduced risk of neuropathy were identified, suggesting potential opportunities for early preventive measures and targets for future drug developments in type 2 diabetic patients.

Introduction section:

The introduction provides a clear overview of the importance and relevance of diabetic neuropathy, emphasizing its prevalence and significant implications for diabetic patients.

Results section:

well-structured and provides a clear presentation of the findings

Consider incorporating visual aids like graphs or figures to enhance the presentation of complex data.

We have added a Manhatten plot of the results that really enhances the presentation of the data (see below).

Figure 1. Manhattan plot showing the logistic regression results of the association analysis.

SNP positions with the corresponding chromosomes are shown at the x axis. P-value as calculated by the logistic regression analysis are presented at the y axis after -log10 transformation. Those SNPs that were considered in this study are denoted by their rsIDs.

Clarify the meaning of " neuropathic symptoms, signs, and parameters " mentioned in Table 3 for better understanding. It might be helpful to clarify what parameters specifically refer to.

Thank you for this suggestion. We have included all the important details in Table 3 as you suggested.

While the genetic variants associated with an increased or decreased risk are well-described, consider adding a brief interpretation or discussion of the clinical implications of these findings. How might these genetic associations inform future diagnostic or therapeutic strategies?

In accordance with the Reviewer's suggestion, we added a brief interpretation to the Discussion section (last but one paragraph).

Discussion Section:

Provide a more detailed interpretation of the functional role of genetic variants and their potential contribution to the pathophysiology of diabetic neuropathy.

We accept the suggestion and have made the appropriate changes.

Explicitly discuss the clinical implications of the findings to help to understand the broader significance of the study in a clinical context. How might the identification of these genetic variants impact the prognosis, diagnosis, or treatment of diabetic neuropathy?

We accept this critical remark and have done our best to improve the quality of the manuscript.

Offer a more detailed comparison with previous studies to highlight similarities or differences in the results.

Thank you for this remark. Since our study is the only whole exome investigation in this field, it is difficult to compare the results of other works with ours. Despite this fact, we tried to do our best to improve the Discussion in this respect, as well.

Acknowledge and discuss any limitations of the study. Are there potential confounding factors or aspects of the research design that could affect the interpretation of results?

We agree and we have added potential coinfounding factors to the last paragraph (limitations) of Discussion.

Finale, suggest potential avenues for future research based on the current findings. Are there specific aspects that warrant further investigation, and how might future studies build upon the current work?

Thank you for this suggestion. We have included these in the Discussion (last but one paragraph).

Methods Section:

Specify the exact criteria used to define neuropathy and include information on normal values or reference ranges for neurological assessments.

We fully agree and we have added two tables (Table 4, 5) which contain the normal values for neurometer and cardiovascular reflex tests.

Elaborate on the thresholds for quality control measures in PLINK v1.9 for transparency.

During the quality control, we have used the recommendations of Marees et al. (Marees, A. T., de Kluiver, H., Stringer, S., Vorspan, F., Curis, E., Marie‐Claire, C., & Derks, E. M. (2018). A tutorial on conducting genome‐wide association studies: Quality control and statistical analysis. International journal of methods in psychiatric research, 27(2), e1608.) for filtering our samples and the detected SNPs. However, our aim was to identify as many variants potentially associated with the development of neuropathy as possible with the number of samples available. Therefore, in some cases, slightly lower thresholds were used, where the recommendations of the PLINK 1.9 software were taken into account. A clarification on the definition of quality parameters was added to the materials and methods section of the manuscript.

Provide additional details on the logistic regression model, explaining why certain variables (age, sex, relatedness) were chosen for adjustment.

By applying the logistic regression, we were interested in the association of certain variants with the development of neuropathy in diabetic patients. However, we have sought to ensure that some factors that might be associated with the phenotype, such as age and sex, do not inadvertently confound our results. Beyond this, in our qualitative analysis we have found a degree of relatedness between our samples that might suggest cryptic relatedness relationships between patients. To eliminate these potential biases, we have adjusted our model for gender, age and estimated relatedness of patients. To further clarify the role of the corrections applied, we have complemented the materials and methods section of the manuscript.

Specify significance levels for Mann-Whitney tests and Fisher’s exact tests used for hypothesis testing.

We would like to thank the Referee for pointing out this mistake. We have supplemented the materials and methods section with information on the p-value threshold used in our analysis.

General Comment:

The cited literature appears to be not up to date, and a further check for more recent literature is advisable. Ensuring that the most current and relevant research is included will strengthen the manuscript's foundation and contextualize the findings within the latest advancements in the field.

We fully agree and we have modified Discussion as well as added the references to the text.

  • Sleczkowska, M.; Almomani, R.; Marchi, M.; de Greef, B.T.A.; Sopacua, M.; Hoeijmakers, J.G.J.; Lindsey, P.; Salvi, E.; Bonhof, G.J.; Ziegler, D.; et al. Peripheral Ion Channel Gene Screening in Painful- and Painless-Diabetic Neuropathy. Int J Mol Sci 2022, 23, doi:10.3390/ijms23137190.
  • Almomani, R.; Sopacua, M.; Marchi, M.; Sleczkowska, M.; Lindsey, P.; de Greef, B.T.A.; Hoeijmakers, J.G.J.; Salvi, E.; Merkies, I.S.J.; Ferdousi, M.; et al. Genetic Profiling of Sodium Channels in Diabetic Painful and Painless and Idiopathic Painful and Painless Neuropathies. Int J Mol Sci 2023, 24, doi:10.3390/ijms24098278.
  • Jankovic, M.; Novakovic, I.; Nikolic, D.; Mitrovic Maksic, J.; Brankovic, S.; Petronic, I.; Cirovic, D.; Ducic, S.; Grajic, M.; Bogicevic, D. Genetic and Epigenomic Modifiers of Diabetic Neuropathy. Int J Mol Sci 2021, 22, doi:10.3390/ijms22094887.
  • Miyashita, A.; Kobayashi, M.; Yokota, T.; Zochodne, D.W. Diabetic Polyneuropathy: New Strategies to Target Sensory Neurons in Dorsal Root Ganglia. Int J Mol Sci 2023, 24, doi:10.3390/ijms24065977.

Once again, we really appreciate your review that has made our manuscript more valuable.

Sincerely, yours,

Dr. Dora Tordai

Round 2

Reviewer 1 Report

Comments and Suggestions for Authors

Many thanks for considering my recommandations

Reviewer 2 Report

Comments and Suggestions for Authors

Thank you for revising the manuscript. The improvements you have made have significantly improved the clarity of your work. All comments have been satisfactorily addressed.